# Boosting Verification of Deep Reinforcement Learning via Piece-wise Linear Decision Neural Networks

**Jiaxu Tian** [1], **Dapeng Zhi** [1], **Si Liu** [2], **Peixin Wang\*** [3], **Cheng Chen** [1], **Min Zhang\*** [1]

[1] Shanghai Key Laboratory of Trustworthy Computing, East China Normal University

[2] ETH Zürich

[3] University of Oxford

51215902014@stu.ecnu.edu.cn, zhi.dapeng@163.com, si.liu@inf.ethz.ch,
peixin.wang@cs.ox.ac.uk, {chchen,zhangmin}@sei.ecnu.edu.cn

## Abstract

Formally verifying deep reinforcement learning (DRL) systems suffers from both inaccurate verification results and limited scalability. The major obstacle lies in the large overestimation introduced inherently during training and then transforming the inexplicable decision-making models i.e., deep neural networks (DNNs), into easy-to-verify models. In this paper, we propose an inverse *transform-then-train* approach, which first encodes a DNN into an equivalent set of efficiently and tightly verifiable linear control policies and then optimizes them via reinforcement learning. We accompany our inverse approach with a novel neural network model called *piece-wise linear decision neural networks* (PLDNNs), which are compatible with most existing DRL training algorithms with comparable performance against conventional DNNs. Our extensive experiments show that, compared to DNN-based DRL systems, PLDNN-based systems can be more efficiently and tightly verified with up to 438 times speedup and a significant reduction in overestimation. In particular, even a complex 12-dimensional DRL system is efficiently verified with up to 7 times deeper computation steps.

## 1 Introduction

Deep neural networks (DNNs) have been exhibiting appealing advantages in decision-making and control for deep reinforcement learning (DRL) systems [1–4]. Nonetheless, the complexity and inexplicability [5, 6] of DNNs render the formal verification of their hosting systems, quite often even themselves, inaccurate and unscalable. Most existing approaches [7–10] over-approximate both embedded DNNs and non-linear environment dynamics to build verifiable models, which inevitably introduces *dual* overestimation. In particular, DNN-specific overestimation is extremely unpredictable due to many factors such as the dimension of system states, the complexity of environment dynamics, and the size, weight, and activation function of a neural network. For example, the verification results may deviate significantly even if the DNNs of the same DRL system differ only in their weights (as we also observed; see Appendix A.4 ). Unsurprisingly, verifying high-dimensional DRL systems would only exacerbate the problems of large overestimation and limited scalability.

Common practice for formally verifying DRL systems is to *train and then transform* the embedded DNNs into easy-to-verify models where, for any input set, output ranges can enclose the outputs of the over-approximated DNNs [7–10]. Taylor models [11] are widely adopted due to their preservation of input-output dependencies and less overestimation (accumulated in multiple steps) than the range analysis approaches such as interval over-approximation [12, 13]. However, they are still prone to intractable overestimation as the accuracy of verification depends heavily on the weights of DNNs

---

\* Corresponding authors.

37th Conference on Neural Information Processing Systems (NeurIPS 2023).

whose effects are difficult to quantify. Several other approaches attempt to extract approximated state-action policies, e.g., decision trees [14, 15], from DNNs via model compression [16] and distillation [17] techniques. However, no equivalence guarantee is established between DNNs and the extracted policies. Consequently, verification results are just probably approximately correct [18].

Inspired by recent advances [19–22] in training near-optimal policies even with reduced training state space imposed by aggregated adjacent states, we propose a novel, inverse *transform-then-train* approach: encoding a DNN into an equivalent set of easy-to-verify linear control policies and *then* optimizing them by training the DNN using reinforcement learning. We accompany our inverse approach by devising a novel neural network model called *piece-wise linear decision neural networks* (PLDNNs), which make linear decisions on each abstract state. Unlike conventional DNNs which build a state-action relation for each actual state, a PLDNN defines a linear relationship, called *Linear Control Unit* (LCU), between actions and actual states associated with the same abstract state. To this end, a PLDNN is essentially a set of LCUs for all abstract states. In contrast to DNNs, LCUs are more explainable and verifiable without any over-approximation. Moreover, PLDNNs are *compatible* with most existing DRL training algorithms as both share the same input and output layers for the same control task.

We extensively assess PLDNN, along with the state-of-the-art tools, with respect to both *performance* (in terms of cumulative rewards and system robustness) and *verifiability* (in terms of overestimation and time cost for the reachability analysis of trained systems) on a collection of benchmarks, including a 12-dimensional control task. Our experimental results show that, compared to the DNN-based systems, the PLDNN-based systems can be verified more precisely, with significantly less overestimation, and more efficiently, with up to 438 times speedup, while achieving comparable performance. Moreover, compared to the state-of-the-art tools, the complex 12-dimensional control task can be trained and verified with up to 7 times deeper computation steps, along with notable tightness improvement.

**Main Contributions.** Overall, we provide: (i) a novel *inverse* approach for boosting the formal verification of DRL systems by learning efficiently and directly (without over-approximation) verifiable piece-wise linear policies with comparable performance; (ii) a novel neural network model to realize the learned piece-wise linear policies, which is compatible with most existing DRL algorithms; and (iii) a prototype called LɪɴCᴏɴ, along with an extensive assessment which demonstrates its tightness in verification results, outperformance over the state-of-the-art tools (up to 438 times speedup), and scalability (up to a 12-dimensional control task).

## 2 Problem Formulation and Motivation

A DRL system is driven by a DNN-implemented controller $\pi$, which is trained for decision-making, and a physical model defined by the ordinary differential equations (ODEs) $\dot{s}(t) = f(s(t), a(t))$, with $s$ the state variables and $a$ a control action. In what follows, we omit the time variable $t$. Typically, for DRL systems, continuous time is discretized, and we have $a = \pi(s)$ at state $s$ and assume $\dot{a} = 0$ during a small time step e.g., $\delta$. At the time point $k\delta$, $k \in \mathbb{N}$, the decision network receives the current state $s_k$ and outputs an action $a_k = \pi(s_k)$. The state variables then evolve according to the physical model during the time interval $[0, \delta]$. The reachable state $s_{k+1}$ at $\delta$ from $s_k$ is $s_{k+1} = s_k + \int_0^\delta f(s, a_k)dt$ which is called the successor state of $s_k$. Note that the system evolves continuously from $s_k$ to $s_{k+1}$. The intermediate states can be computed by substituting the time elapses for $\delta$ in the above formula.

**Definition 1** (Path of DRL systems). *Given a DRL system $\mathcal{D}$ with an environment dynamics $f$, decision network $\pi$, time step size $\delta$ and initial state set $S_0$, a path of $\mathcal{D}$ is a finite or infinite state sequence:* $[s_0, a_0] \xrightarrow[f]{\delta} [s_1, a_1] \xrightarrow[f]{\delta} [s_2, a_2] \xrightarrow[f]{\delta} [s_3, a_3] \xrightarrow[f]{\delta} \cdots$ *such that:*

1. *$s_0 \in S_0$,*
2. *$s_{i+1} = s_i + \int_0^\delta f(s, a_i)dt$ and $a_i = \pi(s_i)$ for $i = 0, 1, 2, \ldots$.*

A DRL system is essentially a DNN-controlled hybrid system. Definition 2 gives a formal definition of regular hybrid systems. The state space of a hybrid system is the Cartesian product of a set $L$ of discrete locations and state space of $n$ real-valued variables $V_c$. At each location $l \in L$, the $n$ continuous variables evolve continuously according to a dynamical law $\dot{v} = f_l(v)$. When the guard condition on the transition between locations $(l_1, l_2) \in T$ is triggered, the system moves to $l_2$, and the continuous variables are reset by $R$. There are two steps involved in the state transition from $(l_i, v_i)$ to its *successor state* $(l_{i+1}, v_{i+1})$: first, from $(l_i, v_i)$ to its *time successor* $(l_i, \varphi_{f_{l_i}}(v_i, t_i))$, and then, to

$(l_{i+1}, v_{i+1})$ that is the *transition successor* of $(l_i, \varphi_{f_{l_i}}(v_i, t_i))$, where $\varphi_{f_{l_i}}$ is the solution of $f_{l_i}$ with initial condition $v(0) = v_i$, mapping the initial state $v_i$ to the state $\varphi_{f_{l_i}}(v_i, t)$ (i.e., the reachable state at time $t$ from $v_i$). Accordingly, the state of a hybrid system can be changed in two ways [23]: (i) by a time delay that changes only the value of continuous variables according to the dynamics of the current location defined in $F$; and (ii) by a discrete and instantaneous transition that changes both location and continuous variables according to the rules in $T$.

**Definition 2** (Hybrid Automata [9]). *A hybrid automaton is an 8-tuple $H = \langle L, Var, Inv, F, T, G, R, I_0 \rangle$, where:*

- *$L$ is a finite set of discrete locations;*
- *$Var$ is a finite set of n real-valued variables with state space $V_c \subseteq \mathbb{R}^n$;*
- *$Inv : L \to 2^{V_c}$ is a function assigning to each location an invariant condition;*
- *$F : L \to (V_c \to \mathbb{R}^n)$ is a function associating each location l to a continuous dynamics $\dot{v} = f_l(v)$;*
- *$T \subseteq L \times L$ is a set of transitions between locations;*
- *$G : T \to 2^{V_c}$ is a function assigning each transition $(l_1, l_2) \in T$ a guard condition $G(l_1, l_2) \subseteq Inv(l_1)$;*
- *$R : T \to 2^{V_c}$ is a function assigning each transition $(l_1, l_2) \in T$ a reset $R(l_1, l_2) \subseteq Inv(l_2)$;*
- *$I_0 \subseteq L \times V_c$ is an initial state set.*

**Definition 3** (Path of Hybrid Automaton [23]). *Let $H = \langle L, Var, Inv, F, T, G, R, I_0 \rangle$ be a hybrid automata. A path of H is a finite or infinite sequence of location and state value pairs, starting from an initial pair $(l_0, v_0) \in I_0$, i.e., $(l_0, v_0) \xrightarrow[f_{l_0}]{t_0} (l_1, v_1) \xrightarrow[f_{l_1}]{t_1} (l_2, v_2) \xrightarrow[f_{l_2}]{t_2} (l_3, v_3) \xrightarrow[f_{l_3}]{t_3} \cdots$ such that:*

1. *$\forall 0 \le t \le t_i, \varphi_{f_{l_i}}(v_i, t) \in Inv(l_i)$*
2. *$(l_i, l_{i+1}) \in T \wedge \varphi_{f_{l_i}}(v_i, t_i) \in G(l_i, l_{i+1}) \wedge v_{i+1} \in R(l_i, l_{i+1})$*

**Theorem 1** (Modeling DRL Systems as Hybrid Automata). *A DRL system with an environment dynamics $f$, decision network $\pi$, time step size $\delta$, and initial state set $S_0$, state space $S$, can be equivalently modeled as the following hybrid automaton:*

- *Var: state variable s, action a, and clock variable $t_c$*
- *$I_0$: $\{(l_0, (s \in S_0, a = 0, t_c = \delta))\}$*
- *L: $\{l_0\}$*
- *Inv: $Inv(l_0) = \{s \in S, t_c \le \delta\}$*
- *F: $F(l_0) = \{\dot{s} = f(s, a), \dot{a} = 0, \dot{t}_c = 1\}$*
- *T: $\{(l_0, l_0)\}$*
- *G: $G(l_0, l_0) = \{t_c = \delta\}$*
- *R: $R(l_0, l_0) = \{t_c = 0, a = \pi(s)\}$*

*Proof.* We first analyze the path of the modeled hybrid automaton $H$. For an arbitrary initial state $(l_0, [s_0, 0, \delta])$, $s_0 \in S_0$, since $t_c = \delta$, the only transition with guard condition $\{t_c = \delta\}$ will be triggered and transition to $(l_0, [s_0, a_0, 0])$ where $a_0 = \pi(s_0)$. Then $(l_0, [s_0, a_0, 0])$ will move to its time successor $(l_0, [s_1, a_0, \delta])$ where $s_1 = s_0 + \int_0^\delta f(s, a_0) dt$. Next, the transition with guard condition $\{t_c = \delta\}$ will be made and conduct the reset operation $t_c = 0, a = \pi(s_1)$ in $R$ to obtain the transition successor $(l_0, [s_1, a_1, 0])$ in which $a_1 = \pi(s_1)$. Repeating the evolution to the time successor and the transition successor yields the following sequence:

$$(l_0, [s_0, 0, \delta]) \xrightarrow[f]{0} (l_0, [s_0, a_0, 0]) \xrightarrow[f]{\delta} (l_0, [s_1, a_1, 0]) \xrightarrow[f]{\delta} (l_0, [s_2, a_2, 0]) \xrightarrow[f]{\delta} \cdots$$

According to Definition 1, there exists the same transition relation $s_{i+1} = s_i + \int_0^\delta f(s, a_i) dt$ with $a_i = \pi(s_i)$ in $\mathcal{D}$ and the modeled hybrid automaton $H$ at each time point $i\delta$. Thus, given an arbitrary initial state $s_0$, the value of state variable and action of $\mathcal{D}$ and $H$ remain the same at each $\delta$. Then with the same state-action pair and dynamics $f$ at each $\delta$, $\mathcal{D}$ and $H$ also have the same state-action value during each time interval $[i\delta, (i + 1)\delta)$. $\qquad\square$

Figure 1 depicts the hybrid automaton defined in Theorem 1. There exists only one location $l_0$. The invariants in $Inv$ claim that any state belongs to the state space $S$ and the clock variable $t_c$ is less than or equal to the time step size $\delta$. The flow in $F$ defines the dynamics $f$ of the system. The only transition is triggered when $t_c = \delta$, updating the action $a$ and resetting $t_c$ as defined in $R$. The continuous change happens during the time interval $[i\delta, (i + 1)\delta]$, with $i \in \mathbb{N}$, and the discrete change of actions occurs at each $\delta$.

Unfortunately, the hybrid automaton of a DRL system cannot be verified by using existing hybrid automata model checkers such as Flow* [24], Ariadne [25], and CORA [26]. The reason is that

the action $a$ in $R$ depends on the uninterpretable DNN $\pi$ by $a := \pi(s)$, and $\dot{s} = f(s, a)$ can not be expressed in a known closed-form, which however is required by regular hybrid automata supported by these tools [27]. Hence, almost all reachability-based verification methods for DRL systems such as Polar [10], Sherlock [7], and ReachNN [28] inevitably over-approximate $\pi$ using a Taylor model, at the cost of large overestimation and time overhead.

## 3 Piece-Wise Linear Control Policies

To bypass the crux of over-approximating DNNs, we devise a novel, alternative neural network model which essentially realizes a set of linear control policies. Our approach bases on the common assumption that there exists a near-optimal linear control policy for every small region of the entire state space [29, 30, 21, 31]. Our objective is then to discretize the state space $S$ of a DRL system and to train a linear control policy for each discretized region. Specifically, given an $n$-dimensional DRL system with $m$-dimensional control input, we train a DNN which implements a linear control function $a_j = b^j + c_1^j x_1 + c_2^j x_2 + \cdots + c_n^j x_n$ for each control dimension $1 \le j \le m$ and each discretized region.

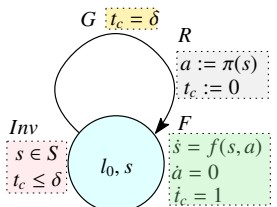

Figure 1: Hybrid automaton for a DRL system.

### 3.1 Abstracting State Space via Abstract Interpretation

Abstract Interpretation [32] is an effective technique for scaling up formal verification of complex systems or programs by reducing the system space while preserving the soundness of verification results. For instance, an infinite state space $[-2, 0] \times [0, 2]$ can be abstracted to be an abstract state represented as $(-2, 0, 0, 2)$, when all the states in $[-2, 0] \times [0, 2]$ share a same property. In general, given a system state space $S$, we denote $S_\phi$ as a finite set of abstract states (each abstract state represents a possibly infinite set of actual system states in $S$). Let $\phi : S \to S_\phi$ be an abstraction function that maps each actual state $s$ in $S$ to the corresponding abstract state in $S_\phi$, and $\phi^{-1} : S_\phi \to 2^S$ be the inverse concretization function such that $\phi^{-1}(s_\phi) = \{s | s \in S, \phi(s) = s_\phi\}$.

For state space abstraction, we choose the very primitive but effective abstraction approach which abstracts actual system states as intervals. It is known as *interval abstract domain* and has been well studied for system [33] and program verification [34] and even the approximation of neural networks [35]. Specifically, let $L_i$ and $U_i$ be the lower and upper bounds for the $i$-th dimension value of $S$. We first define the abstraction granularity as an $n$-dimensional vector $\gamma = (d_1, d_2, \ldots, d_n)$. Then the $i$-th dimension will be divided evenly into $(U_i - L_i)/d_i$ intervals which means each abstract state can be represented as a $2n$-dimensional vector $(l_1, u_1, \ldots, l_n, u_n)$.

**Definition 4** (Interval-Based Abstraction Function). *Given an $n$-dimensional continuous state space $S$ and an abstract state space $S_\phi$ which discretizes $S$ based on abstraction granularity $\gamma$, $\phi : S \to S_\phi$ is called an interval-based abstraction function such that, for every actual state $s = (x_1, \ldots, x_n) \in S$ and abstract state $s_\phi = (l_1, u_1, \ldots, l_n, u_n) \in S_\phi$, we have $\phi(s) = s_\phi$ if and only if $l_i \le x_i < u_i$ holds for each dimension $1 \le i \le n$.*

**Example 1** (Running Example). *Consider a 2-dimensional system in [36] with state space $[-2, 2) \times [-2, 2)$. The dynamics $f$ is defined by following ordinary differential equations (ODE) i.e., $\dot{x}_1 = x_2 - x_1^3$ and $\dot{x}_2 = a$. The sign $a$ means the control action. The objective is to train a DNN for determining action $a$ based on $(x_1, x_2)$ so that the agent can move from the initial region $x_1 \in [0.7, 0.9]$ and $x_2 \in [0.7, 0.9]$ to the goal region $x_1 \in [-0.3, 0.1]$ and $x_2 \in [-0.35, 0.05]$ as soon as possible.*

Suppose that the abstraction granularity is $\gamma = (2, 2)$. The continuous state space $[-2, 2) \times [-2, 2)$ is then partitioned into four regions, corresponding to four abstract states represented by $S_\phi = \{(-2, 0, -2, 0), (-2, 0, 0, 2), (0, 2, -2, 0), (0, 2, 0, 2)\}$, respectively.

### 3.2 Piece-Wise Linear Decision Neural Networks

We devise an alternative DNN model called *piece-wise linear decision neural networks* (PLDNNs). Unlike conventional DNNs, a PLDNN contains an *abstraction layer* between the input layer and the first hidden layer. The abstraction layer is used to convert an actual system state into its corresponding abstract state. Then the output of a PLDNN is the control action that is the *dot product* result of state variables of the actual state and the linear coefficients determined by the corresponding abstract state.

**Algorithm 1:** The Training Procedure based on the DDPG algorithm

1  **Input:** State space $S$, abstraction granularity $\gamma$.     **Output:** A PLDNN $\pi$
2  $\phi \leftarrow$ discretize $S$ according to $\gamma$;        `// obtain abstraction function, Sec3.1`
3  Initialize actor network $\pi$ as a PLDNN by encoding $\phi$ into the coefficient network $\pi_c$; `// Sec3.2`
4  Initialize critic network $Q$, target networks $\pi' \leftarrow \pi, Q' \leftarrow Q$;
5  DDPG($\pi, Q, \pi', Q'$) ;                 `// train` $\pi$ `based on DDPG algorithm`
6  **return** $\pi$

---

Figure 2 exemplifies the architecture of the PLDNN $\pi$ for a two-dimensional DRL system. The decision-making of $\pi$ is based on a coefficient network $\pi_c$ that outputs the linear coefficients. The second layer of $\pi_c$ is the inserted abstraction layer which consists of the blue neurons and the red neurons.

The output layer of $\pi_c$ contains $n + 1$ neurons that output the $n + 1$ linear coefficients depicted as purple neurons. As for the weights setting between the input layer and the abstraction layer, the weights of the connections between the $i$-th neuron in the input layer and the $(2i - 1)$-th and $2i$-th neurons in the abstraction layer are set to 1 which are represented by blue lines and red lines, respectively. While the weights of other connections denoted by the black dashed lines are set to 0. Under this setting of weights, the inputs to both $(2i - 1)$-th and $2i$-th neurons in the abstraction layer are $x_i$. Moreover, the activation function of the $(2i - 1)$-th neuron in

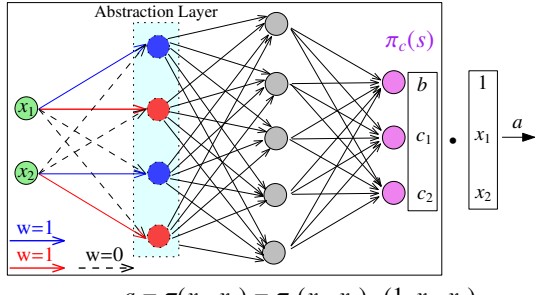

$$a = \pi(x_1, x_2) = \pi_c(x_1, x_2) \cdot (1, x_1, x_2)$$

Figure 2: The arch. of the PLDNN for Example 1.

the abstraction layer is set to $\phi_u$ with the responsibility of computing the upper bound $u_i$, and that of the $2i$-th neuron is set to $\phi_l$ for calculating the lower bound $l_i$. Specifically, for a continuous state space partitioned by the abstraction granularity $\gamma = (d_1, \ldots, d_n)$, the activation functions for the $(2i - 1)$-th and $2i$-th neurons in the abstraction layer can be formulated as follows:

$$\phi_l^i(x_i) = L_i + \lfloor \frac{(x_i - L_i)}{d_i} \rfloor d_i \qquad \phi_u^i(x_i) = L_i + \lfloor \frac{(x_i - L_i)}{d_i} \rfloor d_i + d_i$$

With the above activation functions, the abstraction layer can output the same abstract state $s_\phi = (l_1, u_1, \ldots, l_n, u_n)$ for $\forall s \in \phi^{-1}(s_\phi)$. The abstract state $s_\phi$ is then propagated to the fully connected layers of $\pi_c$ to generate the linear control coefficients $(b, c_1, c_2)$ denoted by $\pi_c(s)$.

To obtain the final output of action $a$, an additional dot product operation between $\pi_c(s)$ and $[1, s]$ is performed with the result of the operation as the control action $a$ where $[\cdot, \cdot]$ is the concatenation operation. For multiple dimensional control action $a = (a_1, \cdots a_m)$, we only need to modify the output dimension of $\pi_c$ to $m(n + 1)$, such that each $n + 1$ neurons output the linear coefficients of one dimension of $a$. More specifically, we can obtain $a_j$ as follows:

$$a_j = \pi(s)_j = \pi_c(s)[(n + 1)(j - 1) : (n + 1)j] \cdot [1, s], 1 \le j \le m.$$

where *vector*[*start* : *end*] denotes the slicing operation that extracts the elements of *vertor* from index *start* up to but not including index *end*.

With the additional abstraction layer that can output an identical vector into the fully connected layers of $\pi_c$ for $\forall s \in \phi^{-1}(s_\phi)$, we can ensure that $\pi_c$ always produces the same coefficients for all actual states located in the same abstract state. Consequently, we can extract a piecewise linear decision function with this structure of $\pi$ on each abstract state.

## 3.3 The Training Procedure

Training a PLDNN can be achieved by extending existing deterministic policy gradient algorithms such as Deep Deterministic Policy Gradient (DDPG) [37] and Twin Delayed Deep Deterministic Policy Gradient [38] since the modifications made stay inside neural networks and are invisible to the DRL algorithms. The pseudo code of the training procedure is given in Algorithm 1, where we take the DDPG algorithm as an example. The training procedure starts with defining the abstraction

function $\phi$ according to $\gamma$ (Line 2), initializing PLDNN with an abstraction layer based on $\phi$ (Line 3), and, following [37], initializing the critic network and the two target networks (Line 4). The procedure then invokes the DDPG algorithm with the networks as arguments (Line 5) since the PLDNN has the same input and output as the actor network implemented by DNN. During this procedure, we freeze the parameters between the input and the abstraction layers of $\pi$. The parameters in the fully connected layers are trained based on backpropagation [39] and gradient descent optimization [40].

## 4 Equivalent Policy Extraction and Verification

After training, we can extract $|S_\phi|$ LCUs based on the learned coefficients of linear control policies for the abstract states in $S_\phi$. Specifically, we choose an arbitrary actual state $s \in \phi^{-1}(s_\phi)$ for each abstract state $s_\phi$ and feed it to a PLDNN to obtain the coefficients defined on the abstract state. For instance, we can feed $(-1, -1)$ to the PLDNN in Example 1 and obtain the coefficients $(-0.16610657, -1.7437580, -1.8227874)$ of the linear control policy for the region $[-2, 0) \times [-2, 0)$. Figure 3 shows the LCUs extracted from a trained PLDNN in Example 1. They are depicted by planes with different colors in Figure 3. These four planes denote the following linear control functions:

$$\pi(x_1, x_2) = -0.16610657 - 1.7437580x_1 - 1.8227874x_2, \quad x_1 \in [-2, 0), \quad x_2 \in [-2, 0) \quad \text{(LCU}_1\text{)}$$
$$\pi(x_1, x_2) = -0.20400035 - 1.8006037x_1 - 1.8679885x_2, \quad x_1 \in [-2, 0), \quad x_2 \in [0, 2) \quad \text{(LCU}_2\text{)}$$
$$\pi(x_1, x_2) = -0.27547930 - 1.8884722x_1 - 1.9342268x_2, \quad x_1 \in [0, 2), \quad x_2 \in [-2, 0) \quad \text{(LCU}_3\text{)}$$
$$\pi(x_1, x_2) = -0.29549897 - 1.9022338x_1 - 1.9436346x_2, \quad x_1 \in [0, 2), \quad x_2 \in [0, 2) \quad \text{(LCU}_4\text{)}$$

The underlying $x_1 \times x_2$ plane in Figure 3 is the projection of the four LCUs. Under the control of these four LCUs, the agent can reach the goal region (orange box). The two sequences of purple boxes represent the range of reachable states from corresponding initial regions to the goal region.

With exacted LCUs from a PLDNN, we can build a verifiable hybrid automaton for the system by substituting equivalently the neural networks using corresponding LCUs. Theorem 2 formulates the hybrid automaton after a decision network $\pi$ is substituted by LCUs. The differences from Theorem 1 include the definitions of transitions $T$, guard condition $G$ and reset formula $R$. For the PLDNN controlled systems, We use $|S_\phi|$ transitions each of which contains a guard condition and a reset formula to update the action $a$ to $\pi_c(s) \cdot [1, s]$ at each $\delta$. Notice that, $\pi_c(s)$ is a determined vector since $\forall s \in \phi^{-1}(s_\phi^i)$, the outputs of $\pi_c$ are the same according to the dedicated structure of $\pi_c$. Thus, the reset formula for $a$ is simplified to an affine mapping.

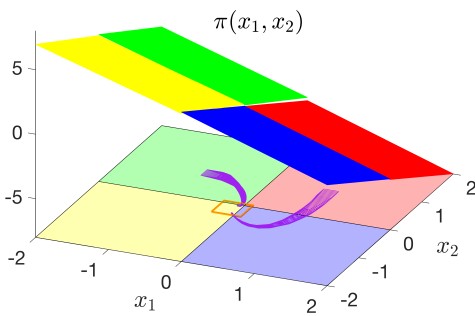

Figure 3: The LCUs extracted from a trained PLDNN for Example 1.

**Theorem 2.** *Given a DRL system with environment dynamics $f$, PLDNN $\pi$, time step size $\delta$ and initial state set $S_0$, it can be equivalently modeled as a hybrid automaton defined as follows:*

- *Var: state variable $s$, action $a$, clock variable $t_c$*  • *$I_0$: $\{(l_0, (s \in S_0, a = 0, t_c = \delta))\}$*
- *L: $\{l_0\}$*  • *Inv: $Inv(l_0) = \{s \in S, t_c \le \delta\}$*  • *F: $F(l_0) = \{\dot{s} = f(s, a), \dot{a} = 0, \dot{t}_c = 1\}$*
- *T: $\{(l_0, l_0), \cdots, (l_0, l_0)\}$ where $|T| = |S_\phi|$*
- *G: $G(T[i]) = \{t_c = \delta, s \in \phi^{-1}(s_\phi^i)\}$ where $0 \le i < |T| \wedge s_\phi^i \in S_\phi$*
- *R: $R(T[i]) = \{t_c = 0, a = \pi_c(s) \cdot [1, s]\}$ where $0 \le i < |T| \wedge s \in \phi^{-1}(s_\phi^i)$*

*Proof.* We prove that the path of hybrid automaton $H_1$ in Theorem 1 is the same as the corresponding hybrid automaton $H_2$ in Theorem 2 for an arbitrary initial state $(l_0, [s_0, 0, \delta])$. The path of $H_2$ can be described by the following sequence:

$$(l_0, [s_0, 0, \delta]) \xrightarrow[f]{0} (l_0, [s_0, a_0, 0]) \xrightarrow[f]{\delta} (l_0, [s_1, a_1, 0]) \xrightarrow[f]{\delta} (l_0, [s_2, a_2, 0]) \xrightarrow[f]{\delta} \cdots$$

where $s_{i+1} = s_i + \int_0^\delta f(s, a_i)dt$ and $a_i = \pi_c(s_i) \cdot [1, s_i]$. With the dedicated structure of PLDNN $\pi$, we have $\pi(s) = \pi_c(s) \cdot [1, s]$. Thus, at each $\delta$, the discrete transitions of $H_1$ and $H_2$ change both location

and continuous variables in the same way. We can conclude that the paths of $H_1$ and $H_2$ are exactly the same. By Theorem 1, it is obvious that $\mathcal{D}$, $H_1$ and $H_2$ produce the same state-action value for the same initial state. □

According to Theorem 2, we can build a hybrid automaton that is equivalent to the DRL system in Example 1. Assuming the trained system has four linear control units as shown in Formulas ($LCU_1$-$LCU_4$) and $\delta = 0.2$, we construct the corresponding hybrid automaton as depicted in Figure 4. The four transitions in the automaton correspond to the four LCUs, respectively. The guard of each transition represents the condition of triggering the corresponding policy.

Thanks to the linearity of control policies in $R$, a hybrid automaton built for a PLDNN-controlled system can be efficiently verified by state-of-the-art tools. For instance, Flow* [24] is a representative tool for the reachability analysis of hybrid systems. In this paper we are focused on the verification of both goal-reach and reach-avoid properties. The former means that given a set of initial states, a system must eventually reach the goal region from any initial state. The latter means that the system never enters unsafe regions within a specific time horizon. Both properties can be verified via reachability analysis.

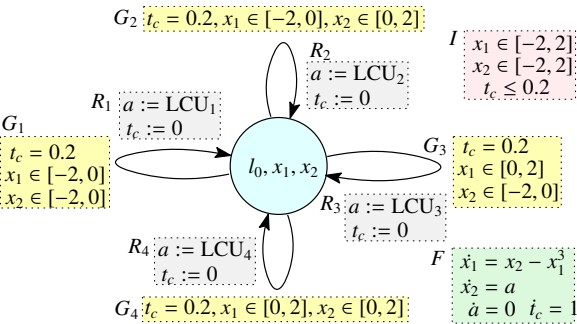

Figure 4: The hybrid automaton of the DRL system with the trained piece-wise linear controllers in Figure 3.

## 5 Experimental Evaluations

We prototype our approach into a tool called LɪɴCᴏɴ, with DDPG as the backend DRL algorithm and Flow* as the verification engine. We extensively assess it, along with the state-of-the-art tools. Our goal is to show, for the PLDNN-based training, (i) the reduction in the number of partitions with comparable cumulative rewards, robustness, and time overhead with respect to conventional DNN-based training; (ii) its high verification performance including the tightness of over-approximation sets and the efficiency of verification; and (iii) its scalability for large-sized neural networks and systems with complex dynamics and high-dimensional state space.

**Experimental Setup.** All experiments were conducted on a workstation equipped with a 32-core AMD Ryzen Threadripper CPU @ 3.6GHz and 256GB RAM, running Ubuntu 22.04.

**Benchmarks.** We choose eight benchmarks, including six regular benchmarks from Verisig 2.0 [9] (B1-B5 and Tora) and two complex benchmarks (CartPole with extreme complex dynamics from OpenAI Gym [41] and quadrotor (QUAD) with 12-dimensional state space and 3-dimensional action space from [10]). For fair comparisons, we use the same training configuration and guarantee that all trained systems reach the specified reward threshold. See Appendix A.1 for the detailed setting. For B1-B5 and Tora the environment dynamics are the same with [9]. The dynamics of QUAD are the same with [10]. We omit their detailed definition here. However, the dynamics of CartPole are originally represented as a discrete-time model in [41] by difference equations. In our experiment, we consider its dynamics as a continuous-time model and formalize it with the following ODEs:

$$\dot{x}_1 = x_2, \quad \dot{x}_3 = x_4,$$
$$\dot{x}_2 = (a + 0.05x_4^2 sin(x_3))/1.1 - (0.05((9.8sin(x_3) - cos(x_3)((a + 0.05x_4^2 sin(x_3))/1.1))/$$
$$(0.5(4.0/3.0 - (0.1cos^2(x_3)/1.1))))cos(x_3))/1.1,$$
$$\dot{x}_4 = 9.8sin(x_3) - cos(x_3)((a + 0.05x_4^2 sin(x_3))/1.1)/0.5(4.0/3.0 - (0.1cos^2(x_3)/1.1)).$$

### 5.1 Performance Evaluation

We assess the performance of PLDNN, together with the conventional DNNs, in terms of cumulative reward, robustness, and training time under the same training configuration. We also measure the number of abstracted states required for training linear control policies and constant policies [21]. Due to space limitations, we present the experimental results only for B1, B2, and two complex cases

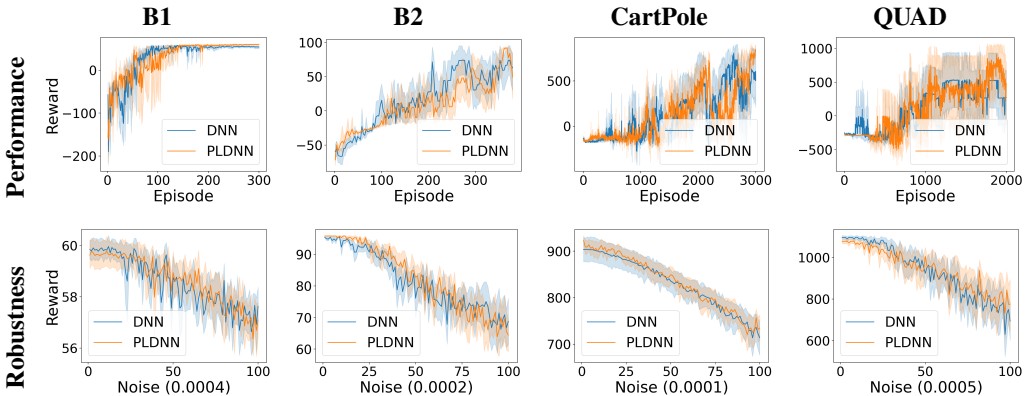

Figure 5: Performance and robustness comparison between PLDNNs and DNNs. The number in the parentheses is the base of $\sigma$, e.g., when the abscissa is 50 in B1, we have $\sigma = 50 \times 0.0004 = 0.02$.

(i.e., CartPole and QUAD). The associated conclusions (from Table 1 and Figure 5) also apply to the other four cases; see Appendix A.2 for the detailed experimental results.

**Cumulative Reward.** Figure 5 plots the system cumulative reward (the average of five trials) during the training process. The solid lines and shadows refer to the average reward and 95% confidence interval, respectively. Apparently, the trends of accumulative rewards by PLDNNs and DNNs

Table 1: Training time and number of partitions

| | Task | B1 | B2 | CartPole | QUAD |
|---|---|---|---|---|---|
| Training Time | PLDNN | 14.3 | 7.9 | 428.2 | 871.1 |
| | DNN | 11.0 | 6.6 | 403.6 | 781.5 |
| #partitions | LCU | 1 | 4 | 16 | 1 |
| | Const. | 4 | 100 | $25^4$ | $4^{12}$ |

are comparable. Despite slightly more time required for training with PLDNN due to its additional conversion from actual state to abstract state (see Table 1, "Training Time"), we can, however, obtain a significant advance in the efficiency and tightness of verification (see Section 5.2).

**Robustness.** We also evaluate the robustness of PLDNNs as training the linear controller on each partition may lead to discontinuity of control decisions in the boundaries of partitions. For a current state $s = (x_1, \ldots, x_n)$, we add a Gaussian noise $X_1, \ldots X_n$ to $s$ and obtain a perturbed state $s' = (x_1 + X_1, \ldots, x_n + X_n)$ for calculating the control action, where $X_i \sim \mathbf{N}(\mu_i, \sigma_i^2)$ with $1 \le i \le n$ and $\mu_i = 0$. For each benchmark, we train 10 different policies and evaluate their robustness under 100 different perturbation levels to obtain the average and 95% confidence interval of the cumulative reward. Figure 5 depicts the reward trend with the increasing perturbation level. As $\sigma$ increases, the decline ratio of the system with PLDNNs is comparable to that with DNNs, which implies that both achieve similar robustness.

**Reduction in the Number of Partitions.** We measure the effect of reducing the number of partitions by utilizing a linear policy, instead of the constant action on each partition, which is adopted by the work [21]. In both cases, we start training from a coarse-grained abstraction granularity (with only one partition) and gradually increase the number of partitions until the preset reward threshold can be reached by the trained PLDNN. As shown in Table 1, linear policies significantly reduce the number of partitions required for reaching the reward threshold, which benefits both the verification efficiency and accuracy as we will see in the next section.

## 5.2 Verification Efficiency and Tightness

We evaluate the verification efficiency and tightness for the PLDNN-based and DNN-based DRL systems, respectively. Regarding tightness, we choose two state-of-the-art tools, i.e., Polar [10] and Verisig 2.0 [9], for the reachability analysis of DNN-controlled systems. We do not consider ReachNN* which both Polar and Verisig 2.0 have been demonstrated to outperform [9, 10]. For efficiency, we employ Flow* to perform reachability analysis, which is used by both Polar and Verisig 2.0 as the backend reachability analysis tool.

**Efficiency.** Table 2 presents the comparison results for the verification efficiency. For each regular case, we choose four different network configurations: two smaller networks (e.g., $Tanh_{2\times20}$) from [10] and two larger networks (e.g., $Tanh_{3\times100}$). Our approach LINCON can handle all 26 instances including the two complex instances, while Polar succeeds only in 20 cases. Verisig 2.0 is not applicable to

Table 2: Verification results and time in seconds.

| Task | Dim | Network | LinCon | | Polar | | | Verisig 2.0 | | | | |
|------|-----|---------|--------|------|-------|------|------|-------------|------|----------|------|------|
| | | | 1 Core | V.R. | 1 Core | Impr. | V.R. | 1 Core | Impr. | 20 Cores | Impr. | V.R. |
| **B1** | 2 | $\text{Tanh}_{2\times20}$ | 2.31 | ✓ | 17 | 7.4× | ✓ | 45 | 19.5× | 38 | 16.5× | ✓ |
| | | $\text{Tanh}_{3\times100}$ | 2.28 | ✓ | 125 | 54.8× | ✓ | 413 | 181.1× | 123 | 53.9× | ✓ |
| | | $\text{ReLU}_{2\times20}$ | 2.11 | ✓ | 3 | 1.4× | ✓ | — | — | — | — | ✗$^c$ |
| | | $\text{ReLU}_{3\times100}$ | 2.59 | ✓ | — | — | ✗$^b$ | — | — | — | — | |
| **B2** | 2 | $\text{Tanh}_{2\times20}$ | 0.57 | ✓ | 5 | 8.8× | ✓ | 5 | 8.8× | 4 | 7.0× | ✗$^a$ |
| | | $\text{Tanh}_{3\times100}$ | 0.56 | ✓ | — | — | ✗$^b$ | — | — | — | — | ✗$^b$ |
| | | $\text{ReLU}_{2\times20}$ | 0.64 | ✓ | 3 | 4.7× | ✓ | — | — | — | — | ✗$^c$ |
| | | $\text{ReLU}_{3\times100}$ | 0.60 | ✓ | — | — | ✗$^b$ | — | — | — | — | |
| **B3** | 2 | $\text{Tanh}_{2\times20}$ | 2.69 | ✓ | 18 | 6.7× | ✓ | 36 | 13.4× | 28 | 10.4× | ✓ |
| | | $\text{Tanh}_{3\times100}$ | 3.57 | ✓ | 91 | 25.5× | ✓ | 357 | 100.0× | 88 | 24.6× | ✓ |
| | | $\text{ReLU}_{2\times20}$ | 3.05 | ✓ | 8 | 2.6× | ✓ | — | — | — | — | ✗$^c$ |
| | | $\text{ReLU}_{3\times100}$ | 2.92 | ✓ | 14 | 4.8× | ✓ | — | — | — | — | |
| **B4** | 3 | $\text{Tanh}_{2\times20}$ | 1.44 | ✓ | 5 | 3.5× | ✓ | 7 | 4.9× | 5 | 3.5× | ✓ |
| | | $\text{Tanh}_{3\times100}$ | 1.45 | ✓ | 27 | 18.6× | ✓ | 114 | 78.6× | 31 | 21.4× | ✓ |
| | | $\text{ReLU}_{2\times20}$ | 1.43 | ✓ | 2 | 1.4× | ✓ | — | — | — | — | ✗$^c$ |
| | | $\text{ReLU}_{3\times100}$ | 1.43 | ✓ | 5 | 3.5× | ✓ | — | — | — | — | |
| **B5** | 3 | $\text{Tanh}_{3\times100}$ | 3.24 | ✓ | 38 | 11.7× | ✓ | 157 | 48.5× | 44 | 13.4× | ✓ |
| | | $\text{Tanh}_{4\times200}$ | 3.29 | ✓ | 157 | 47.7× | ✓ | 1443 | 438.6× | 191 | 58.1× | ✓ |
| | | $\text{ReLU}_{3\times100}$ | 3.28 | ✓ | 7 | 2.1× | ✓ | — | — | — | — | ✗$^c$ |
| | | $\text{ReLU}_{4\times200}$ | 3.29 | ✓ | 49 | 14.9× | ✓ | — | — | — | — | |
| **Tora** | 4 | $\text{Tanh}_{3\times20}$ | 1.57 | ✓ | 45 | 28.7× | ✓ | 69 | 43.9× | 46 | 29.3× | ✓ |
| | | $\text{Tanh}_{4\times100}$ | 1.75 | ✓ | — | — | ✗$^b$ | — | — | — | — | ✗$^b$ |
| | | $\text{ReLU}_{3\times20}$ | 1.58 | ✓ | 30 | 19.0× | ✓ | — | — | — | — | ✗$^c$ |
| | | $\text{ReLU}_{4\times100}$ | 1.62 | ✓ | 53 | 32.7× | ✓ | — | — | — | — | |
| **CartPole** | 4 | $\text{Tanh}_{3\times64}$ | 151 | ✓ | — | — | ✗$^b$ | — | — | — | — | ✗$^b$ |
| **QUAD** | 12 | $\text{Tanh}_{3\times64}$ | 1054 | ✓ | — | — | ✗$^b$ | — | — | — | — | ✗$^b$ |

**Remarks. Impr.**: time speedup of LinCon compared to Verisig or Polar (Verisig or Polar/LinCon).
**Tanh/ReLU**$_{n\times k}$: a DNN with the activation function Tanh/ReLU, $n$ hidden layers, and $k$ neurons per hidden layer. **VR**: verification result. ✓: the reachability problem is successfully verified.
✗$^{type}$: the reachability problem cannot be verified due to *type*: (*a*) large over-approximation error, (*b*) the calculation did not finish, (*c*) not applicable. —: no data available due to ✗$^b$ or ✗$^c$.

ReLU networks (marked by ✗$^c$) and succeeds only in 9 instances. Overall, LinCon outperforms both Polar (up to 47.7× speedup) and Verisig 2.0 (up to 438.6× speedup). In particular, LinCon achieves even up to 58.1× speedup compared to Verisig 2.0 accelerated by 20-core parallelization. For our approach LinCon, the only time overhead for encoding networks stems from extracting LCUs from PLDNNs, which is polynomial and negligible (less than 0.05s). Consequently, LinCon can scale up to large-sized neural networks.

**Tightness.** We compare the tightness of the over-approximation sets computed by different approaches. Figure 6 plots the experimental results, along with the corresponding simulation trajectories. For B2 and Tora, LinCon has a significant tightness improvement over Polar: the range of the over-approximation sets (red boxes) calculated by Polar far outreaches the range of the reachable states obtained from the simulation. Verisig 2.0 terminates prematurely in Tora since the range of the action reaches $10^7$ during the calculation which is too large to proceed. We defer to Appendix A.3 the experimental results for the remaining four cases where all three tools obtain similar tightness results.

**Discussion on CartPole and QUAD.** We discuss the verification results of the two complex cases, namely CartPole and QUAD. Both Polar and Verisig 2.0 fail to verify them. As shown in Figure 6, Polar and Verisig 2.0 abort after 20 steps due to the huge over-approximation error in CartPole. In contrast, such a complex policy trained with our approach can be efficiently and tightly verified. In particular, the trajectories diverge first and finally merge. The computed reachable states tightly over-approximate these trajectories, and the computation takes only 151 seconds. Regarding the 12-dimensional QUAD case, Polar and Verisig 2.0 time out (two hours) after only two steps, while LinCon produces a very tight set of reachable states in 1054 seconds even after 15 steps, which is 7 times deeper than Polar. To the best of our knowledge, this is the first time that QUAD can be formally verified more than ten steps under various decision networks. Note that the trained policies used in the comparison differ due to different decision networks, and thus agents may follow different paths to the goal region (see the simulated trajectories in Figure 6). Hence, for a fair comparison, we conduct more evaluations, from which we can draw the same conclusion as from Figure 6. The results are given in Appendix A.3.

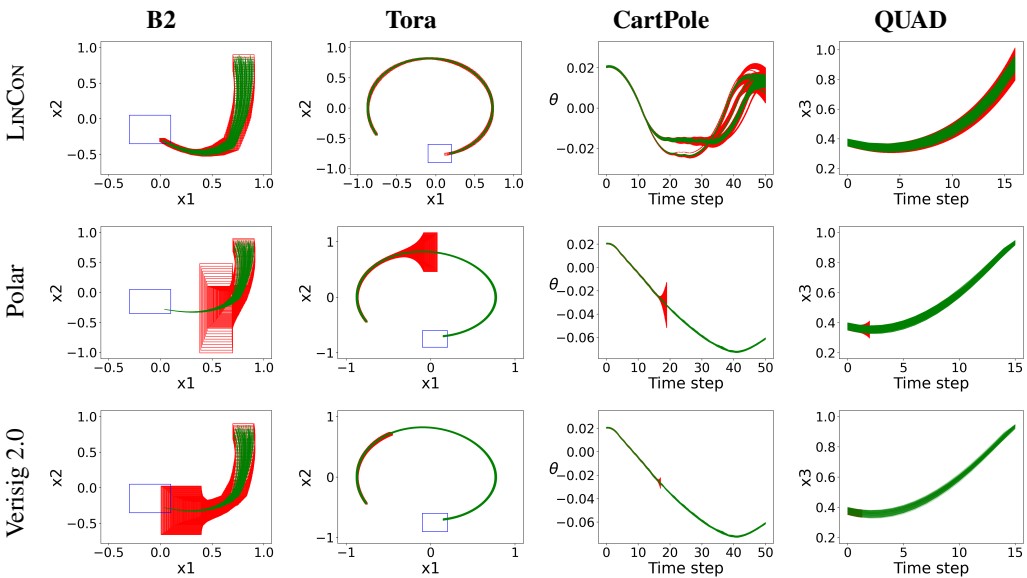

Figure 6: Tightness comparison with respect to the DRL systems with larger decision networks (red box: over-approximation sets; green lines: simulation trajectories; blue box: goal region).

# 6 Related Work

**Policy Synthesis.** Several works adopt programmatic policies (e.g., decision trees and program controllers) which are more interpretable and amenable to formal verification than neural policies. Bastani et al. [15] construct a decision tree to represent a DNN policy based on imitation learning [42]. Verma et al. [43, 44] follow a similar routine, distilling neural network policies into predefined program templates. Trivedi et al. [45] use a two-stage learning scheme to synthesize programmatic policies. Some efforts are dedicated to exploring the combination of training and verification. Zhu et al. [30] propose an inductive framework for synthesizing a deterministic policy program from neural policies. Wang et al. [31] learn programmatic controllers based on verification feedback to avoid safety violations. Our proposed PLDNN is essentially a DNN-based implementation of programmatic controllers, which could be integrated with these verification-guided synthesis approaches.

**Reachability Analysis.** Our work is also built atop the approaches for reachability analysis of neural-network-controlled systems. NNV [46] utilizes star set [47] to perform range analysis of decision networks. JuliaReach [48] uses zonotope propagation to cover the output of a decision network. Verisig [49, 9] models a decision network with differentiable activation functions (e.g., Tanh) as a hybrid system and analyzes its reachability for over-approximating the network. ReachNN* [28, 8] abstracts the input-output mapping of a decision network with a Bernstein polynomial, together with an error bound on the approximation. Sherlock [7] focuses on ReLU-based networks and computes tight Taylor models via rule generation. Polar [10] integrates the Taylor and Bernstein approximation techniques for building a Taylor model which over-approximates decision networks. All these efforts over-approximate the embedded DNNs, which limit their scalability and verification accuracy.

# 7 Conclusion and Future Work

We have presented PLDNN that seamlessly integrates DNN and programmatic controls via state abstraction for boosting the reachability analysis of DRL systems. Unlike traditional train-then-transform approaches, PLDNN accompanies a novel inverse training and verification method, in which a DNN is first transformed into an equivalent set of linear control policies and then trained to optimize them. Experimental results have shown that PLDNN-controlled systems can be more efficiently and tightly verified than DNN-based systems, with up to 438 times speedup and 7 times deeper computation steps for a 12-dimensional control task.

Our work sheds light on a promising direction towards developing dependable DRL systems: learning easy-to-verify and high-performance control policies via DRL and abstraction techniques. Given the encouraging results of linear control policies, our work would also stimulate a passion for substituting them with, e.g., polynomial control policies, for training and verifying more complex DRL systems.

## Acknowledgment

We thank all the anonymous reviewers for their valuable feedback on this work. The work has been supported by the National Key Research Program (2020AAA0107800), NSFC-ISF Joint Program (62161146001,3420/21), Huawei Technologies Co., Ltd., the Shanghai International Joint Lab of Trustworthy Intelligent Software (Grant No. 22510750100), the Shanghai Trusted Industry Internet Software Collaborative Innovation Center, the Engineering and Physical Sciences Research Council (EP/T006579/1), the National Research Foundation (NRF-RSS2022-009), Singapore, and the Shanghai Jiao Tong University Postdoc Scholarship.

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

# A Appendix

## A.1 Benchmarks Setting

We first present the goal region and unsafe region of each benchmark in Table 3.

As for the training settings, for the 6 regular benchmarks, the target is training a policy to guide the agent to reach the goal region. Therefore, we set a negative reward when the agent is not in the goal region. Once the agent reaches the goal region, it will be awarded a positive reward. In CartPole, the target of training is to prevent the pole from falling over, namely the pole angle $-0.2 \leq x_3 \leq 0.2$. We modify the original discrete reward function to a continuous reward function $-|x_3|$ to try to balance the pole to stay upright. As for QUAD, we aim to control the altitude of the quadrotor above 0.6 ($x_3 \geq 0.6$). Thus we set a continuous reward function $-|x_3 - 0.8|$ for training a policy that drives the quadrotor to ascend above 0.6.

Table 3: Benchmarks Setting

| Task | Initial Region | Goal Region | Unsafe Region |
|------|----------------|-------------|---------------|
| B1 | $x_1 \in [0.8, 0.9]$ 
 $x_2 \in [0.5, 0.6]$ | $x_1 \in [0, 0.2]$ 
 $x_2 \in [0.05, 0.3]$ | — |
| B2 | $x_1 \in [0.7, 0.9]$ 
 $x_2 \in [0.7, 0.9]$ | $x_1 \in [-0.3, 0.1]$ 
 $x_2 \in [-0.35, 0.5]$ | — |
| B3 | $x_1 \in [0.8, 0.9]$ 
 $x_2 \in [0.4, 0.5]$ | $x_1 \in [0.2, 0.3]$ 
 $x_2 \in [-0.3, -0.05]$ | — |
| B4 | $x_1 \in [0.25, 0.27]$ 
 $x_2 \in [0.08, 0.1]$ 
 $x_3 \in [0.25, 0.27]$ | $x_1 \in [-0.05, 0.05]$ 
 $x_2 \in [-0.05, 0]$ | — |
| B5 | $x_1 \in [0.38, 0.4]$ 
 $x_2 \in [0.45, 0.47]$ 
 $x_3 \in [0.25, 0.27]$ | $x_1 \in [-0.4, -0.28]$ 
 $x_2 \in [0.05, 0.22]$ | — |
| Tora | $x_1 \in [-0.1, 0.2]$ 
 $x_2 \in [-0.9, -0.6]$ | $x_1 \in [-0.25, 0.10]$ 
 $x_2 \in [0.2, 0.7]$ | — |
| CartPole | $x_1, x_2, x_4 \in [0.02, 0.02]$ 
 $x_3 \in [0.02, 0.021]$ | — | $x_2 \in [26, 29]$ 
 $x_3 \in [-0.2, 0.2]$ |
| QUAD | $x_1, \ldots, x_6 \in [0.35, 0.4]$ 
 $x_7, \ldots, x_{12} \in [0, 0]$ | $x_3 > 0.6$ | — |

## A.2 Comparison on cumulative reward and robustness

In this section, we provide the comparison results on cumulative reward and robustness of B3-B5 and Tora in Figure 7. The solid lines and shadows refer to the average reward and 95% confidence interval, respectively. For these four cases, it is obvious that there are comparable trends in the cumulative rewards of PLDNNs and DNNs during training and under perturbation. Therefore, we can conclude that using PLDNNs will not affect the training efficiency and the robustness of trained systems.

## A.3 Comparison on the tightness of over-approximation sets

**Tightness results of regular cases.** We present the tightness comparison results of B1 and B3-B5 in Figure 8. For these four regular cases, both Polar and Verisig 2.0 can produce tight over-approximation for decision networks, thus all three methods achieve similar tightness results and successfully verify the goal-reach properties.

**Multiple evaluations on CartPole and QUAD.** Since the trained policies used in the comparison differ due to different decision networks, we conduct more evaluations on these two complex cases. For each case, we train three more decision networks with the same configuration and the corresponding results are shown in Figure 10. For CartPole, Polar did not finish the calculation under three different networks as depicted in Figure 10(d-f). Additionally, on the basis of the result of dealing with divergent traces as shown in Figure 10(d), we can see that Polar is not suitable for dealing with the DNN-controlled systems with divergent traces. As for QUAD, we record the computation results within 12 hours and obtain the results as shown in Figure 10(j-l). We can observe that only

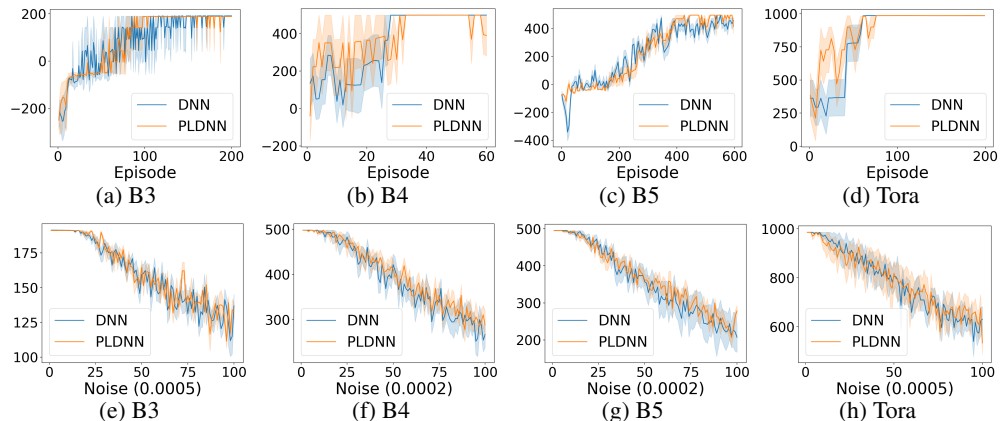

Figure 7: Performance (a-d) and robustness comparison (e-h) of the pldnn and DNN under the same settings. The number in the parentheses is the base of $\sigma$. For example, in B3 when the abscissa is equal to 50, $\sigma = 50 \times 0.0005 = 0.025$.

a few time steps are completed by Polar, while LɪɴCoɴ can accomplish more than 10 steps within about 1000 seconds under multiple evaluations.

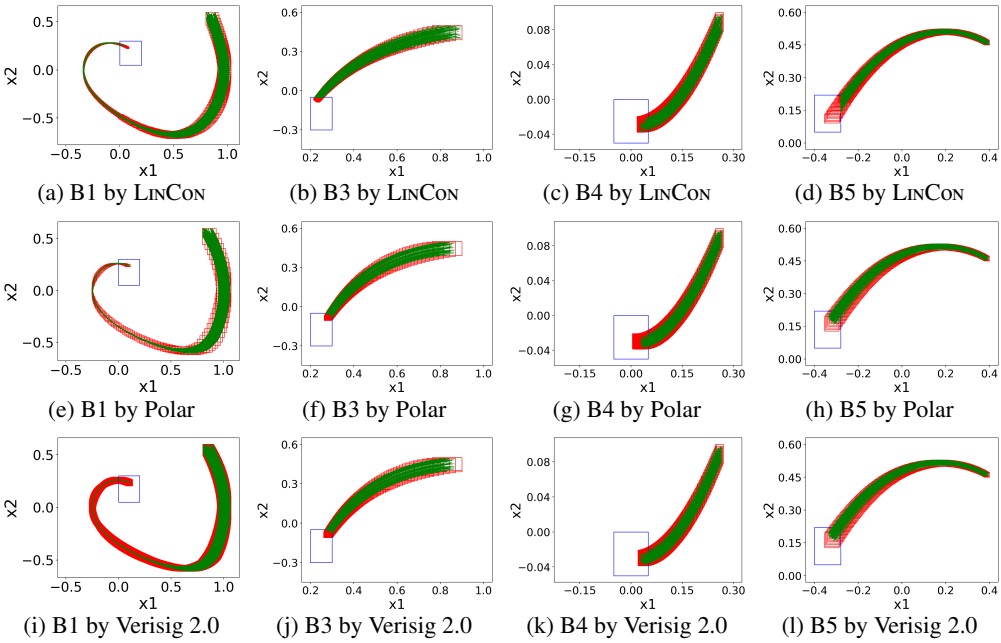

Figure 8: Tightness comparison on the DRL systems with larger decision networks (red box: over-approximation sets; green lines: simulation trajectories; blue box: goal region.)

## A.4 Evaluation on Verisig 2.0 with Big Weights

Verisig 2.0 produces large over-approximation error when dealing with neural networks with big weights. To demonstrate this, we initialize the weights of the neural network with larger values (random numbers $w_l \sim \mathbf{N}(\mu, \sigma^2)$ with $\mu = 0, \sigma = 0.1$) and show the experimental results in Figure 9. We observe that the calculated over-approximation sets contain large over-approximation error except for B4. In Tora, Verisig 2.0 fails to calculate the complete reachable sets due to too large over-approximation error. Hence, it is fairly to say that Verisig 2.0 is sensitive to the DNNs with big weights.

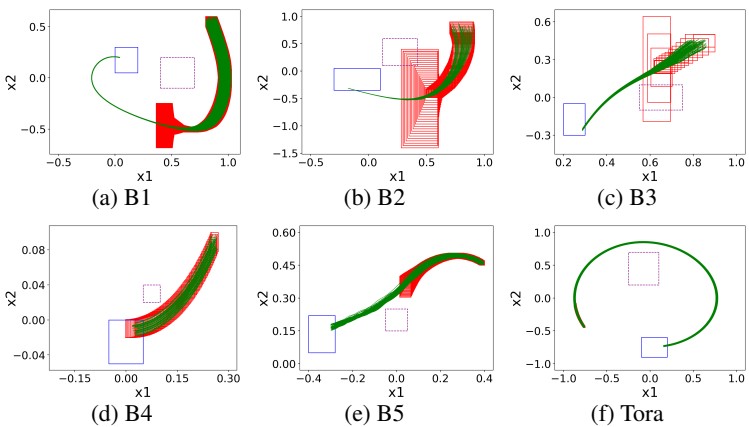

Figure 9: Assessing Verisig 2.0 on the larger networks with big weights. red box: over-approximation set; green lines: simulation trajectories; blue box: goal region; purple dashed box: unsafe region.

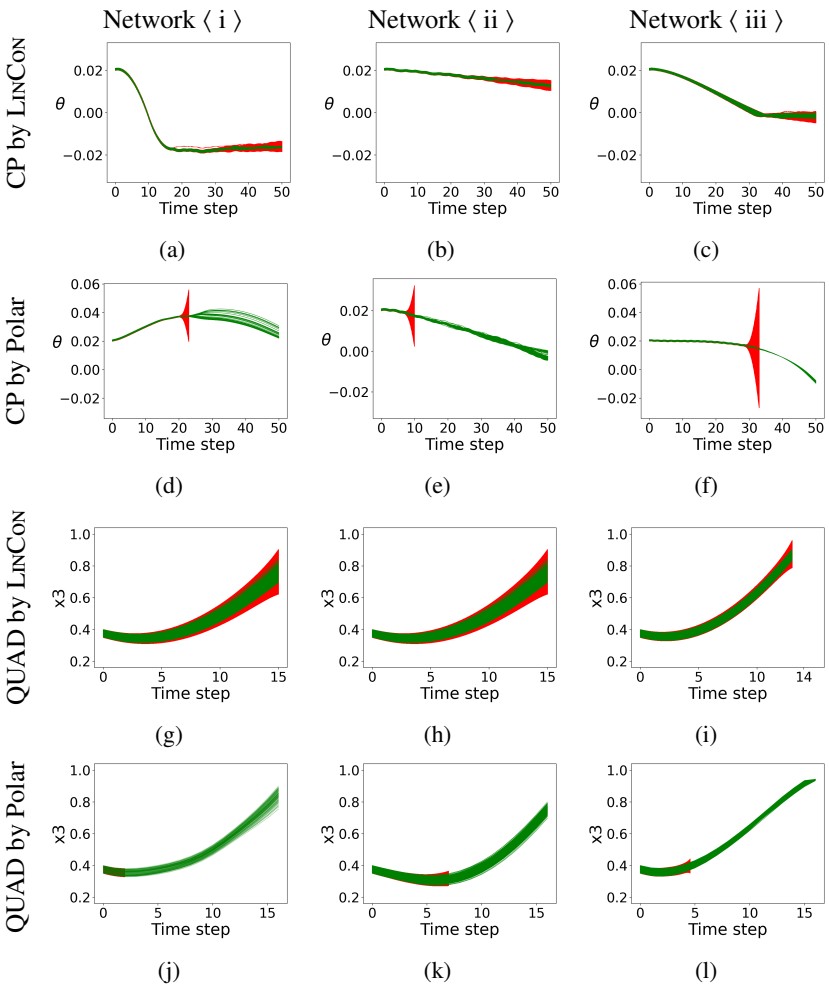

Figure 10: Multiple evaluations on CartPole (CP) and QUAD. red box: over-approximation set; green lines: simulation trajectories;

