# OpenReview forum: "Boosting Verification of Deep Reinforcement Learning via Piece-Wise Linear Decision Neural Networks"
_NeurIPS.cc/2023/Conference — NeurIPS 2023 poster_

### Official Review · Reviewer_eCb4 · 2023-06-20

**Soundness:** 3 good
**Presentation:** 2 fair
**Contribution:** 2 fair
**Rating:** 6
**Confidence:** 4

**Summary:**

This paper proposes an inverse transform-then-train approach for verifying deep reinforcement learning systems. It encodes a DNN into efficiently verifiable linear control policies and optimizes them via reinforcement learning. The approach is compatible with existing DRL training algorithms and shows that PLDNN-based systems can be more efficiently and tightly verified, with up to 438 times speedup and a significant reduction in overestimation.


**Strengths:**

+ The paper explores an interesting and important direction
+ Innovative and practical methodology

**Weaknesses:**

- Some technical parts are unclear
- The way of presenting contributions of the work is somewhat misleading
- The limitation of using PLDNN is not discussed

**Questions:**

##
## Originality

To efficiently verify deep reinforcement learning systems, this paper proposes an inverse transform-then-train technique. It efficiently converts a DNN into verifiable linear control policies, which it then uses reinforcement learning to improve. The method is compatible with current DRL training algorithms and demonstrates that PLDNN-based systems can be more effectively and tightly verified, with speedups of up to 438 times and a notable decrease in overestimation.

##
## Importance of contribution

The idea of using PLDNN to boost the verification of DRL is interesting. However, it seems like it converts the DNN into a different structure before training, and the PLDNN is different from the original DNN. To me, it is more like a synthesis technique, rather call it as verification. Moreover, the authors claim that it is compatible with most existing DRL training algorithms, but in the evaluation, the details of the training algorithms are not shown.

##
## Soundness

This paper proposes an inverse transform-then-train approach for verifying deep reinforcement learning systems, overcoming inaccuracies and scalability issues. It uses piece-wise linear decision neural networks (PLDNNs) for efficient verification, reducing overestimation. However, there are some technical parts that are not so clear and need further clarification.

Page 1, Line 21
> Most existing approaches [7–10] over-approximate both embedded DNNs and non-linear environment dynamics to build verifiable models, which inevitably introduces dual overestimation.

I am not sure what this dual-estimation is. The sentence following seems only talk about the disadvantage of over-approximation.

Page 2, Line 40
> ..., we propose a novel, inverse transform-then-train approach:

It seems like the approach cannot verify a trained DRL agent. So it is more like a synthesis technique.

Page 5, Line 197
> Consequently, we can extract a piecewise linear decision function with this structure of π on each abstract state.

How the piecewise linear functions are extracted? Could you explain it more detailedly?

##
## Evaluation

The authors try to evaluate whether PLDNN-based training offers reduced partitions, comparable rewards, robustness, time overhead, high verification performance, and scalability for large-sized neural networks and complex systems with high-dimensional state space. In Section 5.2, the authors compare Lincon with polar and verisig 2.0. Have you compared the efficiency of these tools on the original DNNs? Since, maybe the architecture of PLDNN is not easy to be handled for Polar and Verisig 2.0.

##
## Quality of presentation

The paper is not that easy to follow. I would recommend the authors add more motivations for the design of each single component.

##
## Comparison with Related Work

The PLDNN structure is similar to the DNN with abstract states listed below. What are the main difference and advantages of using PLDNN?

- Jin, Peng, et al. "Trainify: a CEGAR-driven training and verification framework for safe deep reinforcement learning."

**Limitations:**

No, the authors did not discuss the limitation of the work. I think more attention should be given to the disadvantages of using PLDNN, like is it still the same original network?

---

> ### Author Rebuttal · Authors · 2023-08-09
>
> **Question for the importance of contribution:** (i) It is more like a
> synthesis technique, rather call it verification. (ii) Algorithm
> compatibility of PLDNN.
>
> **Response:** (i) Yes, it can be understood as a synthesis method for
> developing verification-friendly models. We show that the powerful
> fitting capability of DNN can be leveraged to train easy-to-verify
> PLDNNs. One novelty of our work is that we use DNN as a backbone to
> implement policy synthesis. (ii) Our approach is compatible with most
> existing DRL training algorithms because we do not change the input and
> output structure of the decision network. Namely, the differences with
> the original DNN are invisible to the DRL algorithms.
>
> **Question for Page 1, Line 21:** What this dual-estimation is?
>
> **Response:** The first overestimation is introduced when dealing with
> the DNN controller. For the traditional DNN, it cannot be expressed in a
> known closed-form expression. Consequently, most approaches resort to
> computing a conservative model such as a Taylor model that encloses the
> output of DNN given an input set, which is the first overestimation for
> the decision neural network. The second overestimation is introduced to
> obtain an overestimation model for the solution of ODE system dynamics.
> That is unavoidable as most nonlinear ODEs do not have closed-form
> solutions. Using PLDNN as the decision neural network, we can eliminate
> the first overestimation as PLDNN can be expressed in a known
> closed-form (i.e., piece-wise linear function).
>
> **Question for Page 2, Line 40:** It seems like the approach cannot
> verify a trained DRL agent. So it is more like a synthesis technique.
>
> **Response:** Yes, the traditional DNN-based agent cannot be verified
> directly in our approach. Our insight is that training/synthesizing
> verification-friendly models is more practical to develop certified DRL
> systems than directly verifying the complex canonically trained DNNs. We
> demonstrate this in the paper.
>
> **Question for Page 5, Line 197:** How the piecewise linear
> functions are extracted?
>
> **Response:** The extraction is straightforward. Due to the special
> design of PLDNNs (Figure 2 in our submission), we can feed an arbitrary
> state in each partition into the embedded coefficient network $\pi_c$
> and it will output the linear coefficients of the corresponding linear
> controller defined on each partition. As partitions are finite, we can
> traverse all the partitions and obtain the extracted linear controllers
> that are equivalent to the PLDNN.
>
> **Question for Evaluation:** Have you compared the efficiency of
> these tools on the original DNNs? Since, maybe the architecture of PLDNN
> is not easy to be handled for Polar and Verisig 2.0.
>
> **Response:** It is true that PLDNNs cannot be handled by Polar and
> Verisig 2.0, but that is not our intention in this work. As we can
> extract equivalent linear controllers from a PLDNN, we do not need to
> cope with the network decision model for verification. Instead, we can
> leverage the off-the-shelf hybrid system verification tools such as
> Flow\* to verify PLDNN-based systems. To demonstrate the
> verification-friendliness, we provide the verification results by PLDNN
> and original DNNs in Table 2 in our submission, respectively. The
> results show that DNN-based systems are difficult to verify even by
> using state-of-the-art tools like Polar and Verisig 2.0 and hardware
> acceleration. In contrast, PLDNN-based systems are more efficient to
> verify even without hardware acceleration. Moreover, the verification
> results are more precise as the DNN over-approximation is avoided.
>
>  **Question for Comparison with Related Work:** What are the main
> differences and advantages of using PLDNN?
>
> **Response:** There are two main advantages of our approach. First, we
> train a linear controller on each partition while other approaches such
> as Trainify [1] train a constant action. Consequently, the number of
> partitions we need is far less than that needed by Trainify. Second,
> with PLDNN, we can solve the verification problem by transforming the
> PLDNN-based system into an equivalent hybrid model, which can be
> verified using off-the-shelf tools such as Flow\*. However, Trainify
> needs to build Kripke structures from the trained models, which, quite
> often, suffers from the notorious state explosion problem due to the
> exponential increase of partitions during training.
>
> [1] P. Jin, J. Tian, D. Zhi, X. Wen, and M. Zhang, “Trainify: A cegar-driven training and verification framework for safe deep reinforcement learning,” in International Conference on Computer Aided Verification. Springer, 2022, pp. 193–218. 10

---

> > ### Comment · Reviewer_eCb4 · 2023-08-11
> >
> > Thank you for the answer. I would like to change the score to weak accept.

---

### Official Review · Reviewer_63JN · 2023-07-05

**Soundness:** 4 excellent
**Presentation:** 3 good
**Contribution:** 4 excellent
**Rating:** 7
**Confidence:** 3

**Summary:**

The paper presents an approach for designing neural network policies, in the context of deep reinforcement learning (DRL) with continuous state and action spaces, that are more amenable to verification compared to standard networks. A standard approach to verifying continuous DRL systems is to abstract the system after learning the policy and to then verify the abstracted system. In contrast, the proposed approach first abstracts the state space (such that the state space is discretized) and then learns a linear policy for each abstract state. The coefficients of the linear policy are learnt using a coefficient network that maps each abstract state to its resulting policy coefficients. This mapping itself need not be linear and can be a complex neural network. However, since the state space is discretized and the policy for each abstract state is fixed and linear, tight and efficient verification of the resulting DRL system becomes feasible. Moreover, the state abstraction computation is itself encoded as the first layer of the coefficient network, so that the resulting policy network (referred to as piece-wise linear decision neural networks or PLDNNs) has a standard input (system state) and an output (actions), and can be directly trained using off-the-shelf DRL algorithms. The empirical evaluation shows that the performance of PLDNNs, measured in terms of cumulative reward and robustness of the policy, is comparable to standard policy networks. At the same time, DRL systems with PLDNN policies are much more amenable to verification.

**Strengths:**

Even when the system dynamics are known, formal verification of safety and liveness properties of DRL systems is challenging to scale, specially for continuous state and action spaces. The paper proposes a very interesting idea for architecting policy networks that are easier to verify. I believe that the proposed notion of abstracting the state space before learning the policy will be a fruitful research direction. I also find the observation that a small set of linear policies can perform as well as a complicated non-linear policy to be surprising and interesting in its own right. I should add the disclaimer that I am not an expert on the topic of reinforcement learning and do not have a good sense of the broader literature on the topic.

**Weaknesses:**

My main concern is about the "scalability of verification vs cumulative reward" tradeoff for DRL systems with high-dimensional state spaces (for instance, when the inputs to the policy network are images). As the state space becomes more complex, I suspect that a finer-grained partition becomes necessary to learn a good policy, directly impacting the scalability of verification. The presented evaluation considers system states with up to 12 dimensions, so it is hard to make an assessment about PLDNNs in settings where inputs are images required hundreds to thousands of dimensions.
I am also curious about how the proposed approach would be used when the action space is discrete. Would each abstract state be mapped to a fixed action? Some discussion about this setting could be interesting and helpful. Finally, I have minor concerns about the empirical evaluation (described in the Questions section below).

**Questions:**

I have some clarification questions about the evaluation.

1. What were the model architectures used for the results reported in Section 5.1? Are these the same architectures as in Section 5.2?

2. Are the architectures for PLDNN and DNN the same (except for the first layer) for the results in Section 5.1 and Section 5.2? If not, how were the DNN architectures chosen and how do we ensure that it is an apples-to-apples comparison?

3. Is the time reported in Table 1 in seconds? And is it the total training time over all episodes?

4. How is the reward for the robustness experiments calculated, i.e., what is the horizon length? More generally, what is the reward function? The appendix gives some details; it says that for the 6 regular benchmarks "we set a negative reward when the agent is not in the goal region. Once the agent reaches the goal region, it will be awarded a positive reward". How and what is the positive reward assigned? Why is the final cumulative reward so different across the 6 regular benchmarks (based on Fig 5 and 7) if their reward functions are so similar?

5. Are the models used for the robustness experiment the same as the ones used for the cumulative reward experiment? If yes, then why do you say that 10 different policies are trained for the former while you conduct five trials for the latter? My understanding is that the models used for the robustness experiment are trained on non-perturbed data but the evaluation is conducted with perturbed data. If this is so, then the same models can and should be used for both sets of experiments.

6. What are precisely the properties of the DRL systems being verified? Based on Table 3 in the appendix, my guess is that for B1-B5, Tora, and QUAD the verified property is a liveness property whereas for CartPole it is a safety property? Can you please confirm? Is the time horizon for properties bounded or unbounded? It would be helpful to include this detail in the paper as well.

Some more general questions:

1. Can you comment on the applicability of this approach in settings where the inputs are high-dimensional images?

2. Can the approach be adapted to settings where the action space is discrete?

**Limitations:**

The paper does not necessarily discuss the limitations and might benefit from a small discussion about the same.

---

> ### Author Rebuttal · Authors · 2023-08-09
>
> **Weaknesses 1:** (i) It is hard to make an assessment about PLDNNs in
> settings where inputs are images. (ii) How the proposed approach would
> be used when the action space is discrete?
>
> **Response:** (i) Yes, applying PLDNNs when inputs are images is
> difficult due to the high dimensionality. At present, we are only
> focused on state-based cases. One potential solution to adapting PLDNNs
> to images is first to extract features from images and train PLDNNs on
> the extracted features. We believe that this is an interesting research
> direction to follow.
>
> \(ii\) For discrete action space, please refer to our response to
> **General Question 2** below.
>
> **Question 1:** What were the model architectures used for the
> results reported in Section 5.1? Are these the same architectures as in
> Section 5.2?
>
> **Response:** We use the larger network structure with Tanh activation
> function (e.g. Tanh$_{3 \times 100}$), as shown in Table 2 in our submission, to conduct
> the training and robustness evaluation.
>
> **Question 2:** Are the architectures for PLDNN and DNN the same
> (except for the first layer) for the results in Section 5.1 and Section
> 5.2?
>
> **Response:** Yes, they are almost the same. Precisely, they have the
> same architectures except that PLDNN has an abstraction layer and an
> additional output layer for outputing linear coefficients. The hidden
> layers of PLDNN and DNN are exactly the same in our comparison
> experiments.
>
> **Question 3:** Is the time reported in Table 1 in seconds? And is
> it the total training time over all episodes?
>
> **Response:** Yes, the time reported in Table 1 in our submission is in seconds, and it is
> the total training time over all episodes.
>
> **Question 4:** How is the reward for the robustness experiments
> calculated, i.e., what is the horizon length? reward function? Why is
> the final cumulative reward so different across the 6 regular
> benchmarks?
>
> **Response:** The horizon length in CartPole is set to 200. In the other
> benchmarks, the settings of horizon length depend on the number of
> control steps needed to reach the goal region. For example, in B1, it
> takes about 30 control steps to reach the goal region after the training
> phase. In B2, the agent needs about 15 control steps to reach the goal
> region. Based on this, we set the horizon length slightly larger than
> the control steps, e.g. 35 for B1 and 20 for B2.
>
> For the reward function, the positive reward when the agent reaches the
> goal region is set to a constant such as 100. The negative reward
> function is set according to the distance between the agent and the
> center of the goal region. For instance, since the goal region of B1 is
> $x_1 \in [0,0.2], x_2 \in [0.05,0.3]$, we set the negative reward
> function as $-|x_1 -0.1| - |x_2 - 0.2|$. Hence, to obtain a higher
> cumulative reward, the agent needs to reach the goal region as soon as
> possible.
>
> As different control steps are needed to reach the goal region in each
> benchmark, the cumulative reward is different across the six regular
> benchmarks even if their reward functions are similar.
>
> **Question 5:** Are the models used for the robustness experiment
> the same as the ones used for the cumulative reward experiment?
>
> **Response:** Yes, the five models used in the reward evaluation are the
> same as for the robustness evaluation. However, as the robustness
> assessment is based on perturbed data, which may cause uncertainties, we
> have therefore performed five additional trials to produce more precise
> averaged robustness evaluation results.
>
> **Question 6:** (i) What are the properties being verified? (ii) Is
> the time horizon for properties bounded or unbounded?
>
> **Response:** (i) For B1-B5, Tora, and QUAD, we verify a liveness
> property that checks whether the agent can reach the goal region within
> some bounded control steps. For CartPole, we verify a safety property:
> whether the agent can stay in the safe region within some specific
> control steps. (ii) Yes, all the time horizons are bounded because our
> verification is based on reachability analysis which only calculates the
> overestimated sets of reachable states in finite steps, as required by
> the off-the-shelf tool $\text{Flow}^*$.
>
> **General Question 1:** Can you comment on the applicability of this
> approach in settings where the inputs are high-dimensional images?
>
> **Response:** Since PLDNN-based training only costs slightly more time
> than traditional DNN-based training, we believe that PLDNNs can be
> applied to the training phase when dealing with high-dimensional image
> input such as image classification. That is, the piece-wise linear
> function $\pi$ and softmax function $\mathit{softmax}$ are composed to
> yield a function that outputs the probability of each category in the
> form of $\mathit{softmax}(\pi(s))$. However, the verification cannot
> directly handle the high-dimensional image input due to the extremely
> high-dimensional input space. Another reason is that the dynamics of
> pixels may not be definable. If we can extract some informative features
> from the image input first, combining the verification technique may be
> feasible.
>
> **General Question 2:** Can the approach be adapted to settings
> where the action space is discrete?
>
> **Response:** Yes. One straightforward approach is to make each abstract
> state perform the same action. Moreover, if we still want to maintain a
> linear output on each abstract state, we only need to generate a mapping
> between the continuous output and the discrete action space. For
> example, given a discrete action space $\{1,-1\}$, when the output of
> the decision neural network is greater than 0, the agent executes action
> 1; otherwise, it performs action -1:
>
> $$a=
> \begin{cases}
>     1 &  \text{if }\pi(s)>0,\\\\
>     -1 & \text{otherwise.}
> \end{cases}$$

---

> > ### Comment · Reviewer_63JN · 2023-08-15
> > **Response to rebuttal**
> >
> > Thank you for the detailed response to my questions. I will keep my score.

---

### Official Review · Reviewer_uQsi · 2023-07-06

**Soundness:** 3 good
**Presentation:** 3 good
**Contribution:** 3 good
**Rating:** 5
**Confidence:** 4

**Summary:**

The paper proposed a new neural network architecture, PLDNN, for better verification of DRL trained / neural-network controlled closed-loop controlled systems. PLDNN differs by abstracting the input space with intervals and applying linear mappings in each abstract states. The controller represented by PLDNN can be integrated with different DRL algorithms. The experimental results showed that the PLDNN can retain the performance by training with DRL algorithms directly and achieve tighter reachable sets with better verification efficiency.

**Strengths:**

- The paper is sound and the topic of the paper is of high interest to the research community.
- The paper does a good job at introducing technical details and makes the paper easy to follow.
- The empirical study shows great improvement of the proposed method over existing reachability analysis methods for neural-network controlled systems.

**Weaknesses:**

- My main concern of the paper is lack of analysis on the policy of reducing the partitions. Though the experimental results show strong results of PLDNN, it is not clear how partitioning plays a role in the performance improvement. It could be for some partitioned space in the state space, a linear control policy is almost close to the optimal control policy. Some other partitioned region may need finer partitions to approach the optimal control policy with a combination of linear control policies. However, similar insights or observations are not provided in the paper. Some analysis on how number of partitions evolves during training or a comparison between the linear policy of partition reduction and a baseline, like a fixed number of partition, would be great to have.
- The paper positioned PLDNN as a new transform-then-train approach. Training with PLDNN may result in some implicit benefits of finding near-optimal policy represented by PLDNN. But PLDNN and the partition schema are still applicable for the train-then-transform process, e.g., distilling a trained DNN to PLDNN. I would recommend authors to provide an empirical comparison between using PLDNN with transform-then-train and train-then-transform processes.

**Questions:**

- Could you provide more details about the linear policy for reducing the partitions?
- Could you provide more details on how network are trained for benchmarks in the experimental section, e.g., training time? Are all network trained from scratch?

**Limitations:**

Some limitation discussion on training time cost would be great to have.

---

> ### Author Rebuttal · Authors · 2023-08-09
>
> **Weaknesses 1:** My main concern of the paper is lack of analysis on
> the policy of reducing the partitions. Though the experimental results
> show strong results of PLDNN, it is not clear how partitioning plays a
> role in the performance improvement. Some analysis on how number of
> partitions evolves during training or a comparison between the linear
> policy of partition reduction and a baseline, like a fixed number of
> partitions, would be great to have.
>
> **Response:** From the perspective of function fitting, more partitions
> tend to obtain a policy that fits the optimal decision function better,
> which implies a better reward performance. In our experiment, we compare
> with an approach which trains constant actions on partitioned regions
> [1]. Under a similar performance constraint, the
> partition reduction is significant by our approach (Table 1 in our submission). Using a
> fixed number of partitions is a reasonable baseline which can also show the
> effect of partition reduction. In our future work, besides the baseline with a fixed number of partitions, we will further
> consider building another baseline, i.e., the minimal number of
> partitions that are needed for training near-optimal piece-wise linear
> controllers.
>
> **Weaknesses 2:** I would recommend authors to provide
> an empirical comparison between using PLDNN with transform-then-train
> and train-then-transform processes.
>
> **Response:** Thanks for your insightful suggestion!
> Train-then-transform via PLDNN is indeed a direction that deserves
> further study. We did not consider this as one of our intentions is to
> demonstrate the feasibility of training verification-friendly and
> near-optimal PLDNNs directly. We agree that PLDNNs are also achievable
> via the train-then-transform approaches. Encouraged by the suggestion,
> we carried out a quick experiment on the comparison. The results show
> that the directly trained PLDNN exhibits similar performance (reward is
> 52.97) to a canonically trained DNN (reward is 52.45), while there is
> a small decrease in performance (reward is 51.25) of the PLDNN
> transformed from the trained DNN. See the attached PDF file and our
> global response for more details. However, such a decrease is almost
> negligible and more comprehensive experiments are required to draw fair,
> conclusive results.
>
> **Question 1:** Could you provide more details about the linear
> policy for reducing the partitions?
>
> **Response:** We compare the partition reduction of linear policy with
> an approach [1] that trains a constant action on each
> partition (Table 1 in our submission). For both approaches, we start from a coarse-grained
> partition (i.e., one region) to train using the DDPG algorithm and
> increase the number of partitions on each dimension until the preset
> reward threshold is achieved. The linear policy relaxes the constraint
> that each abstract state needs to correspond to a constant action. Our
> evaluation shows that using linear policy, rather than a constant, can
> significantly reduce the number of partitions under the same reward
> threshold setting, e.g., from $25^4$ to $16$.
>
> **Question 2:** Could you provide more details on how networks are
> trained for benchmarks in the experimental section, e.g., training time?
> Are all networks trained from scratch?
>
> **Response:** We use DDPG as the training algorithm for all PLDNNs and
> DNNs. In addition, all the networks are trained from scratch and the
> corresponding average training time is reported in Table 1 in our submission (in seconds).
> The setting of the reward functions is given in Appendix A.1 of our
> accompanying technical report (submitted as supplementary material). We
> use the larger network structure with the Tanh activation function
> (Table 2 in our submission) to conduct the training and robustness evaluation.
>
> [1] P. Jin, J. Tian, D. Zhi, X. Wen, and M. Zhang, “Trainify: A cegar-driven training
> and verification framework for safe deep reinforcement learning,” in International
> Conference on Computer Aided Verification. Springer, 2022, pp. 193–218.
> 10

---

> > ### Comment · Reviewer_uQsi · 2023-08-15
> > **Response to rebuttal**
> >
> > Thank you for the detailed response especially for the newly added analysis on train-then-transform approaches. I would encourage the authors to investigate how transform-then-train may benefit the reward compared with the train-then-transform in the future work. At the current stage, I would like to keep my original score.

---

### Official Review · Reviewer_DN9m · 2023-07-14

**Soundness:** 3 good
**Presentation:** 3 good
**Contribution:** 2 fair
**Rating:** 6
**Confidence:** 4

**Summary:**

This paper presents an approach towards more easily verifiable DRL agent. Instead of training a neural network and then applying verification tools to it, the paper proposes to partition the input state, train linear policies in each of the partition, and verify the resulting piecewise-linear policy as a hybrid automation in Flow*.

**Strengths:**

- Verifying DRL systems has gained research attention in the past several years due to its supposed application in many safety/cost-critical scenarios. Currently, scalability is a major concern. Therefore, the topic of the paper is of interests to the ML and verification community.

- While most existing work verifies a neural network controller via input splitting + abstract interpretation, the paper proposes an interesting and novel alternative, which is to eagerly partition the input region before training and make sure that a linear policy is trained for each partition. Compared to the train-and-transform approach, the proposed approach guarantees linearity in each partition, which leads to faster verification empirically.

- The proposed method is relatively easy to implement in existing training framework. The engineering trick that inserts hand-crafted neural network layers to make sure input from the same partition will be multiplied with the same weights is clever.

**Weaknesses:**

Conceptually, the method leverages the observation that a relatively small set of linear functions is sufficient to achieve comparable performance to a more complex neural network. This observation is itself rather surprising and deserve a closer study. For the same input region where the PLDNN is linear, is the behavior of the neural network also relatively linear? Are there input region where non-linearity is truly needed and PLDNN's behavior is problematic?

If low training loss can truly be obtained by a set of linear functions, it seems reasonable to expect that the behavior of the neural network in the same partition is close to linear as well, making abstract interpretation based techniques relatively precise. Following this thought,  the performance gain in verification precision might be a construct of the baseline neural network being unnecessarily large. Have the authors tried whether smaller networks can result in similar performance?


**Questions:**

How do you determine the number of segments along each dimension?

How is the system dynamic handled? Is it encoded precisely?

To my knowledge, Verisig 2.0 also uses Flow*. Is it fair to say that the main difference between LinCon and Verisig 2.0 is that the former verifies a PLDNN while the latter verifies a canonically trained neural network?

**Limitations:**

The paper does not explicitly discuss its current limitation.

---

> ### Author Rebuttal · Authors · 2023-08-09
>
> **Weaknesses 1:**\
> (i) The observation that a relatively small set of linear functions is
> sufficient to achieve comparable performance deserves a closer study.\
> (ii) For the same input region where the PLDNN is linear, is the
> behavior of the neural network also relatively linear?\
> (iii) Are there input region where non-linearity is truly needed and
> PLDNN's behavior is problematic?
>
> **Response:** (i) We hold the same opinion and believe that this
> observation would stimulate more studies in this research direction. In
> fact, there have recently been several related works. For example,
> in [1], a deterministic program with a small set of
> linear functions (less than four) is learned as a safety shield. In
> [2], the "if-then-else\" structured programs whose
> depth of syntax tree is between two and five can fulfill different
> control tasks. Hence, it is feasible to achieve good performance with a
> relatively small set of linear functions.
>
> \(ii\) The answer is yes if there is a unique optimal control policy.
> This is because the PLDNN can be considered as an approximation of the
> optimal decision neural network. However, if there can be multiple
> optimal control policies, an agent may make different decisions even in
> the same state with different policies, but all the different decision
> sequences can achieve optimal performance globally. In this case, the
> linearity of a PLDNN in the same input region does not necessarily imply
> that the neural network has the same relatively linear behavior.
>
> \(iii\) Yes, in theory, regions may exist where non-linearity behavior
> is truly needed. However, a nonlinear function can be well-fitted by a
> set of linear functions from the perspective of function fitting. The
> widely-used ReLU neural networks are such an example, which are
> essentially a piece-wise linear function as well (see
> [3] for details). More partitions can be applied to
> perform a similar non-linear decision.
>
> **Weaknesses 2:** Can smaller networks result in similar
> performance?
>
> **Response:** Yes, that is possible in practice. However, whether
> smaller networks result in similar performance depends on the complexity
> of DRL control tasks, such as the system dynamics and the control
> target. Our experimental results show that: for some simple cases such
> as B1 in Table 2 in our submission, the smaller networks Tanh$(2 \times 20)$ and
> ReLU$(2 \times 20)$ can achieve comparable performance. In contrast, in
> CartPole with relatively complex system dynamics, a small neural network
> such as Tanh$(2 \times 20)$ can not obtain high performance.
>
> **Question 1:** How do you determine the number of segments along
> each dimension?
>
> **Response:** The number of partitions depends on the training reward of
> the DRL control system. We start training from a small number of
> partitions. If the preset reward threshold is not reached, meaning that
> there may exist some regions that need a non-linear decision, we will
> further divide the state space till the preset reward threshold can be
> reached.
>
> **Question 2:** How is the system dynamic handled? Is it encoded
> precisely?
>
> **Response:** Yes, the system dynamics $f$ is exactly encoded into the
> flow component of a hybrid automaton in the form of ordinary
> differential equations (ODEs)
> $F(l_0): \{\dot{s} = f(s, a), \dot{a} = 0, \dot{t}_c = 1\}$. We then
> use the tool Flow\* to compute conservative results that contain the
> solution of the ODEs.
>
> **Question 3:** Is it fair to say that the main difference between
> LinCon and Verisig 2.0 is that the former verifies a PLDNN while the
> latter verifies a canonically trained neural network?
>
> **Response:** Yes, that is the main difference. Our insight is that
> training verification-friendly controllers such as PLDNNs can be a good
> alternative to developing certified DRL systems. As shown in our
> experiments, the verification results by LinCon are very close to the
> simulation results, while the results by Verisig 2.0 may fail the
> verification (Table 2 in our submission) due to the over-approximation of canonically
> trained neural networks.
>
> [1] H. Zhu, Z. Xiong, S. Magill, and S. Jagannathan, “An inductive synthesis frame-
> work for verifiable reinforcement learning,” in Proceedings of the 40th ACM SIG-
> PLAN conference on programming language design and implementation, 2019, pp.
> 686–701.
>
> [2] Y. Wang and H. Zhu, “Verification-guided programmatic controller synthesis,” in
> International Conference on Tools and Algorithms for the Construction and Analysis
> of Systems. Springer, 2023, pp. 229–250.
>
> [3] G. F. Montufar, R. Pascanu, K. Cho, and Y. Bengio, “On the number of linear re-
> gions of deep neural networks,” Advances in neural information processing systems,
> vol. 27, 2014.
> 9

---

> > ### Comment · Reviewer_DN9m · 2023-08-15
> >
> > Thank you for the detailed response. I would like to change my score to 6.

---

### Author Rebuttal · Authors · 2023-08-09

## Discussion on Limitations
We thank all the reviewers for the valuable feedback. We first briefly
discuss the main limitations of our method, as raised by all the
reviewers.

One limitation concerns a potential rapid increase in the number of
partitions when a preset reward threshold can never be reached. One
solution would be to locate and partition the regions with poor
performance. These regions that result in the failure of training can be
regarded as *counterexamples* and should be further divided. We plan to
leverage the counterexample-guided abstraction and refinement (CEGAR)
formal method to cope with this problem.

Another potential limitation is that the verification complexity may
still be high for piece-wise linear controllers, although we show in the
paper that they are more amenable and verification-friendly than neural
network controllers. At present, we employ off-the-shelf hybrid
verification tools [1] to demonstrate the effectiveness of
our approach. We are considering implementing dedicated algorithms to
improve the verification efficiency.

## Train-Then-Transform using PLDNN

Encouraged by Reviewer 63JN's suggestion, we have conducted a quick
experiment to compare the two approaches: transform-then-train and
train-then-transform. The experiment was conducted on B1. The results
are presented in Table 1 in our PDF file.

**Experimental Setting.** We use the same network structure
Tanh$_{2 \times 20}$ for training the following three decision networks
(i.e. DNN, PLDNN, and Distilled PLDNN). For DNN and PLDNN, the training
algorithm is DDPG. The distilled PLDNN is obtained using supervised
learning in which the training data is obtained through sampling from
the traces generated by the DNN controller.

In the train-then-transform approach, we first train a DNN controller
whose cumulative reward is about 52.45. Then we use supervised learning
to distill the DNN's policy into a PLDNN containing four partitions.

**Results.** Regarding performance, i.e., the cumulative reward, the
directly trained PLDNN can reach 52.97, which is similar to that of the
canonically trained DNN (52.45). We also observe that there is a slight
decrease in the system performance after the DNN is distilled to a PLDNN
(51.25).

Regarding verification, both the distilled PLDNN and the directly
trained PLDNN can be verified faster than the DNNs, which demonstrates
PLDNN's verification-friendliness. Moreover, the verification result of
the directly trained PLDNN is more precise, i.e., tightly closer to the
simulation results. Both the distilled PLDNN and original DNN result in
larger overestimation due to the over-approximation.

**Conclusion.** The preliminary experimental results show that, compared
to our transform-then-train approach, the train-then-transform approach
can achieve similar verification performance, but may cause a
performance decrease and extra overestimation of verification results.
Nevertheless, our comparison is preliminary; more comprehensive
experiments are therefore needed to draw a fair conclusion.

[1] X. Chen, E.  ́Abrah ́am, and S. Sankaranarayanan, “Flow*: An analyzer for non-
linear hybrid systems,” in Computer Aided Verification: 25th International Confer-
ence, CAV 2013, Saint Petersburg, Russia, July 13-19, 2013. Proceedings 25. Springer,
2013, pp. 258–263.

---

### Decision · Program_Chairs · 2023-09-21

**Decision:**

Accept (poster)

**Comment:**

The paper gives a new approach to learning verified RL policies, in which a neural policy is transformed into a collection of verifiable linear control policies, which are then optimized using RL.

The idea that a small number of verifiable linear controllers can collectively achieve comparable performance to a neural policy -- at least in some interesting domains -- is very intriguing. While the reviewers originally had some concerns about the evaluation, these concerns were addressed during the discussion period. Given all this, I am delighted to recommend acceptance. Please carefully incorporate the reviewers' feedback in the final version.